# The Molecular Mechanism of Polyphenols in the Regulation of Ageing Hallmarks

**DOI:** 10.3390/ijms24065508

**Published:** 2023-03-14

**Authors:** Quélita Cristina Pereira, Tanila Wood dos Santos, Isabela Monique Fortunato, Marcelo Lima Ribeiro

**Affiliations:** 1Laboratory of Immunopharmacology and Molecular Biology, Sao Francisco University Medical School, Braganca Paulista 12916-900, SP, Brazil; 2Lymphoma Translational Group, Josep Carreras Leukemia Research Institute, 08916 Badalona, Spain

**Keywords:** ageing hallmarks, polyphenols, molecular mechanisms

## Abstract

Ageing is a complex process characterized mainly by a decline in the function of cells, tissues, and organs, resulting in an increased risk of mortality. This process involves several changes, described as hallmarks of ageing, which include genomic instability, telomere attrition, epigenetic changes, loss of proteostasis, dysregulated nutrient sensing, mitochondrial dysfunction, cellular senescence, stem cell depletion, and altered intracellular communication. The determining role that environmental factors such as diet and lifestyle play on health, life expectancy, and susceptibility to diseases, including cancer and neurodegenerative diseases, is wellestablished. In view of the growing interest in the beneficial effects of phytochemicals in the prevention of chronic diseases, several studies have been conducted, and they strongly suggest that the intake of dietary polyphenols may bring numerous benefits due to their antioxidant and anti-inflammatory properties, and their intake has been associated with impaired ageing in humans. Polyphenol intake has been shown to be effective in ameliorating several age-related phenotypes, including oxidative stress, inflammatory processes, impaired proteostasis, and cellular senescence, among other features, which contribute to an increased risk of ageing-associated diseases. This review aims to address, in a general way, the main findings described in the literature about the benefits of polyphenols in each of the hallmarks of ageing, as well as the main regulatory mechanisms responsible for the observed antiageing effects.

## 1. Introduction

The ageing process is a natural process, accompanied by a progressive decrease in physiological activities, a fact that can culminate in compromised metabolic functions, increased physical vulnerability, and consequently, a greater risk of death. These factors have been associated with the development of chronic diseases such as cancer, diabetes, obesity, osteoporosis, osteoarthritis, cognitive decline, dementia, disability, and heart diseases, as well as several neurodegenerative diseases [1].

In recent years, several studies have shown that there is a strong influence of epigenetic events in controlling the ageing rate. These findings describe a set of common players in ageing, known as genomic instability, telomere attrition, epigenetic changes, loss of proteostasis, dysregulated nutrient sensing, mitochondrial dysfunction, cellular senescence, stem cell exhaustion, and altered intercellular communication [1,2,3,4]. Currently, strategies to delay the ageing process involve telomere reactivation, inflammation control, mTOR inhibition with calorie restriction or the use of bioactive compounds, and sirtuin activation, among others [1,5].

In light of such options, polyphenol-rich compounds maybe an interesting strategy because they have well-characterized antiageing properties. Moreover, they are widely found in nature, such as polyphenols from blueberries, catechins from green tea, theaflavins from black tea and procyanidins from apples, resveratrol, curcumin, and epigallocatechin gallate (EGCG), among others. Accordingly, several studies have revealed that polyphenols can modulate several important phenomena in the ageing process. These phytochemicals have a diverse pharmacological profile as well as connections to a wide variety of biological targets [6,7,8,9,10,11,12].

The beneficial effects of polyphenols include the control of the redox state of cells, modification of cell signalling, and protection against damage to biological molecules such as nucleic acids, lipids, and proteins. These effects can occur both directly through the elimination of reactive oxygen species (ROS) and indirectly through interaction with transcription factors that modulate the antioxidant response. Additionally, it has been demonstrated that polyphenols can increase the expression of antioxidant enzymes such as superoxide dismutase (SOD) and catalase [13]. Moreover, some studies have shown that curcumin, resveratrol, and quercetin have a protective role against oxidative stress-induced damage both in vitro and in vivo [14,15].

Polyphenols can also decrease the inflammatory response, modulate nutrient sensing pathways, and induce selective apoptosis of senescent cells. These biological events are important promoters of disease development, as they become dysfunctional with advancing age [16,17]. Therefore, the precise elucidation of the molecular mechanisms of polyphenols in the modulation of biological phenomena related to ageing becomes challenging due to the complexity of biological systems, where diverse biochemical aspects contribute to the establishment of this phenotype [18].

The availability of public information regarding the potential benefits of natural products such as polyphenols for human health has led to increased consumption of these substances in a wide variety of forms, such as nutritional supplements, fortified foods, or even fresh foods rich in these compounds [5]. However, some studies have found that excessive consumption of polyphenols may have negative health effects in some population groups, and therefore recommend that caution regarding the consumption of polyphenol-enriched products and supplements should be considered to ensure safe intake levels. In addition, consumption of foods naturally rich in polyphenols in general should be encouraged and prioritized, given their innumerable beneficial effects and because the risk of high doses of polyphenols in these natural sources is low [5,19,20].

Therefore, the achievements reached in recent decades concerning the identification of ageing markers have greatly favoured the development of research aimed at understanding the regulatory mechanisms exerted by polyphenols on these molecular events, as well as bringing new perspectives for the use of these substances to delay the ageing process [5,21,22]. Therefore, this review will address the relevance of the use of polyphenols and their possible effects on the modulation of the main markers of ageing described in the literature.

## 2. Age-Related Effects of Polyphenols on Epigenetic Changes

Epigenetic changes comprise one of the hallmarks of ageing and are an important mechanism involved in the process of deterioration of cellular functions observed during ageing [2]. The discovery of epigenetic mechanisms has made significant contributions to clarifying complex facts such as the difference in ageing patterns between identical twins [23]. The information encoded by the epigenome includes DNA methylation, chromatin remodelling, posttranslational modifications of histones, and transcription of noncoding RNAs (ncRNAs). The combination of these different types of epigenetic information determines the function and fate of all cells and tissues [24]. Epigenetic changes play a vital role in the healthy development of an organism because they are crucial for various biological processes, such as transcription, cell division, and DNA replication. Therefore, the overall stability of epigenetic mechanisms is crucial for maintaining proper molecular activity, which reduces the possibility of various diseases and contributes to the slowing down of the ageing process [25].

The influence of environmental factors, such as exercise and diet, is known to have a direct impact on gene expression and longevity in different organisms [26,27]. Some evidence points to the beneficial effects of polyphenolic compounds on ageing-related events, including anticancer and anti-inflammatory effects. Diet plays a key role in the regulation of epigenetic modifications, such as DNA methylation and DNA demethylation, and histone modifications regulated by enzymes such as histone deacetylases (HDACs), histone acetyltransferases (HATs), and histone methyltransferases (HMTs) [27,28,29]. In this regard, the effect of diallyl disulphide (DADS), an organosulphur compound found mainly in garlic, on HDAC activity and the regulation of gene expression in the human colorectal cancer cell lines Caco-2 and HT-29 has been shown. DADS treatment induced histone H3 acetylation in both Caco-2 and HT-29 cell lines, as well as hyperacetylation of histone H4 at lysine 12 and lysine 26 in Caco-2 cell lines. Moreover, DADS treatment also increased the expression of the cell cycle regulator p21Waf1/Cip1 at the mRNA and protein levels in both cell lines [30].

EGCG, the main bioactive compound found in green tea, has been intensively studied for its demethylating properties by acting as a DNA methyltransferase (DNMT) inhibitor in several types of lung cancer, leukaemia, and breast cancer, as well as in some neurodegenerative disorders. In a study performed in a cell model of nonsmall cell lung cancer (NSCLC), the A549 cell line was treated with EGCG prior to cisplatin (DDP) treatment. The results of this study showed that EGCG administration in A549/DDP cell lines resensitized cells to DDP and resulted in inhibition of cell proliferation, cell cycle arrest in G1 phase, and increased apoptotic activity. In addition, EGCG treatment inhibited DNMT and HDAC activity, which led to a decrease in the expression of growth arrest specific 1 (GAS1), tissue inhibitor of metalloproteinase 4 (TIMP4), and intercellular adhesion molecule 1 (ICAM1) genes [31]. Similarly, another study performed with human breast cancer MCF-7 and leukaemia HL60 cell lines indicated that EGCG administration resulted in diminished cell proliferation and induced apoptosis in both cell lines. Furthermore, EGCG treatment leads to hypomethylation in the hTERT promoter region and inhibition of histone 3 lysine 9 (H3K9) acetylation in MCF-7 cells [32]. EGCG extracted from green tea was shown to reduce 5-methylcytosine, mRNA, and protein levels of DNMT1, DNMT3a, and DNMT3b in human epidermoid carcinoma A431 cells. In addition, EGCG treatment also decreased HDAC activity and increased the acetylation of lysine 9 and 14 on histone H3 [33]. A further study also suggested that EGCG inhibits DNMT activity, which may reactivate the expression of genes silenced by methylation in human colon cancer (HT-29), oesophageal cancer (KYSE-150), and prostate cancer (PC3) cell lines [34].

Genistein, another polyphenol present mainly in olives, soybeans, and fava beans, has been studied for its antiageing properties. Considering that renal fibrosis is a common histomorphological feature of renal ageing, it has been shown that treatment with genistein mitigates renal fibrosis in mice. Mechanistically, it was found that polyphenols can restore the activity of the klotho promoter, an antiageing protein that is abundant in renal tissue and suppresses fibrosis. In addition, simultaneous inhibition of histone 3 deacetylation and DNMT1 and DNMT3a activity was also observed [35].

Curcumin is another polyphenol that is wellknown for its anti-inflammatory, antiproliferative, and anticancer properties [36,37]. It was demonstrated in the human breast cancer MCF-7 cell line that curcumin was able to inhibit DNMT1 activity. In addition, curcumin treatment also led to reactivation of Ras-associated domain family protein 1A (RASSF1A) and further decreased cell proliferation and tumour growth [38]. Another study investigating the potential of curcumin showed that the treatment increased RA receptor beta (RARβ) gene expression and decreased tumour growth and DNMT3b activity in the human lung cancer cell line A549 [39]. Considering that DNA methylation is an important event involved in the genesis of several ageing-related diseases, evidence of the relationship between dietary polyphenols and DNA methylation may provide new strategic insights of paramount importance for healthy ageing. Table 1 summarizes the main effect of polyphenols in this hallmark.

Because epigenetic deregulation is a recurrent event in the ageing process, this evidence endorses the role of dietary polyphenols and their impact on the modulation of this event, bringing new perspectives to studies on this subject, as well as allowing us to understand the mechanism’s epigenetic changes (Figure 1) as a molecular target related to dietary changes and longevity.

## 3. Effects of Polyphenols on Genomic Instability

The accumulation of genome damage and somatic mutations leading to genome instability are important drivers and hallmarks of ageing [2,67,68]. Genomic instability is associated with age-related diseases such as cancer, heart failure, type 2 diabetes, chronic obstructive pulmonary disease, stroke, Alzheimer’s disease, Parkinson’s disease, chronic kidney disease, atherosclerosis, osteoporosis, and sarcopenia [69,70].

Single-cell genome sequencing and transcription sequencing have made it possible to trace the somatic mutational landscape of the human body, including age-dependent dynamics [71,72,73]. In this regard, it has been observed that throughout the ageing process, the frequency of DNA damage and somatic mutations in animal and human tissues increases, resulting in genomic instability, which is expressed through the generation of point mutations, breaks, DNA strand cross-linking, transpositions, translocations, and aneuploidies [67]. Importantly, different somatic cells accumulate mutations at different rates, which results in the formation of cell clones with a slightly different genotype in an ageing organism, culminating in somatic mosaicism [74,75,76]. This phenomenon is extremely widespread even among healthy people [77,78].

There are several levels of cellular protection against DNA damage and the accumulation of mutations. These mechanisms involve the elimination of DNA-damaging molecules, repair of DNA damage, and elimination of dysfunctional cells through senescence and apoptosis. In addition, the maintenance of chromatin structure, especially constitutive heterochromatin, plays an important role in ensuring the integrity and stability of genome function [70,79,80,81]. In young organisms, compensatory mechanisms are activated to prevent phenotypic and functional changes. However, increased stress and impaired functioning of these mechanisms with age lead to the accumulation of damage, exceeding the functional threshold [82]. The dysregulation of these pathways can lead to accelerated or premature ageing, a decline in the functional capacity of vital organs, and the development of age-related diseases [83].

In this sense, the antioxidant defence system is one of the most important mechanisms to prevent damage to cellular macromolecules. Oxidative stress leads to an age-related increase in the cellular level of oxidatively modified macromolecules, including DNA, and this increase is associated with various pathological conditions such as ageing, carcinogenesis, and neurodegenerative and cardiovascular diseases. This condition is counteracted by the antioxidant defence system, which includes enzymatic and nonenzymatic processes [84]. In ageing organisms, the activity of antioxidant enzymes is significantly decreased, while levels of free radicals and oxidative damage to DNA are increased [85,86].

An important aspect strongly associated with the maintenance of genome stability is associated with dietary habits. It is well established that dietary elements play an essential role in nucleotide synthesis and DNA replication, maintenance of DNA methylation and chromosome stability, prevention of DNA oxidation, and recognition and repair of DNA damage [87,88]. In addition, polyphenolic compounds such as curcumin, EGCG, resveratrol, and quercetin have been shown to reduce the level of DNA damage and stimulate the DNA damage response, including the regulation of sensors, transducers, and mediators [12,27,83]. In this regard, it was demonstrated that resveratrol can induce an increase in the genome stability of mouse embryonic fibroblasts, which protects the cells against the induction of mutations in the ARF/p53 pathway. In addition, replication stress-associated DNA double-strand breaks that accumulated with genomic destabilization were effectively reduced by treatment with polyphenol [59]. Other studies have suggested that proanthocyanidins and their microbial metabolites can increase the expression of DNA repair genes and activate the ATM and ATR proteins [60,89,90]. It is important to highlight that the inactivation of proteins involved in the DNA damage response process has been described in several age-dependent diseases, including cancer, as well as progeroid syndromes [83]. Table 1 summarizes the main effect of polyphenols in this hallmark.

In this sense, the genomic and geroprotective effects exerted by polyphenols on genomic instability are evident. These results can be achieved through different mechanistic actions, such as prevention of oxidative damage in DNA, elimination of free radicals, reduction inDNA double-strand breaks and DNA adduct formation, among others, giving these compounds a promising role in the regulation of ageing markers (Figure 2).

## 4. Effects of Polyphenols on Telomere Attrition

Telomere shortening is a well-known mechanism linked to ageing. The decrease in efficiency of the replicative process of telomeres over the years results in considerable shortening of this region with each generation of the cell until it reaches a critical length [91,92]. At this stage, the speed of cell division slows markedly, triggering replicative cell death. However, differences in cell type determine a longer or shorter period of survival and preservation of the metabolism of these cells. Cells involved in growth, development, and reproduction, such as stem cells, eggs, and sperm cells, synthesize large amounts of telomerase, an enzyme involved in the preservation of telomeric DNA length. On the other hand, most adult cells express this enzyme little, or even do not express it, thus favouring ageing and subsequent cell death [93,94]. It has been proposed that oxidative stress and free radicals play a crucial role in telomere shortening by inhibiting both telomerase activity and telomere repeat binding factor 2 (TERF2) expression levels. The shortening of telomere length leads to genomic instability and consequent impairment of cell cycle function, senescence, and apoptosis of cells, resulting in health harm [15,95].

In addition, critically short telomeres cause responses to DNA damage, defective mitochondrial biogenesis, and negative regulation of sirtuins, linking telomeres to metabolic control [96,97,98]. An association between increased oxidative stress and inhibition of SIRT1 expression/activity, which results in senescence related to telomere shortening, has been pointed out [15,99]. Recently, a new axis involving short telomere metabolism NAD-SIRT1-mitochondrial ROS has been reported and supports the global effects of short telomeres [100,101].

Currently, in an attempt to increase telomerase activity, maintain telomere length and extend lifespan, the use of antioxidant supplements such as polyphenols has been extensively investigated [102]. It has been proposed that polyphenols, with their antioxidant and anti-inflammatory capabilities, may impact telomere length and effectively prevent telomere shortening. The effects of dietary antioxidants on telomere function indicated that diet is a significant factor in determining the status of telomere length. In this regard, it has been shown that leukocyte telomere length can be considerably improved in individuals on a Mediterranean diet rich in olive oil [103]. Furthermore, it has been described that the use of polyphenols such as proanthocyanidins and procyanidins, found in grape seed extract, with recognized antioxidant and anti-inflammatory potential, allowed for adecrease in apoptosis and prevention of hydrogen peroxide-induced chromosomal damage in human lymphoblastic cells [52]. A further study showed that EGCG and quercetin were able to prevent apoptosis of cardiac myocytes by preventing telomere shortening and loss of TERF2 expression, the results that were attributed to the potential antioxidant effect of these compounds [104]. A cross-sectional study including men and women in China associated the preservation of telomere length in elderly Chinese men with the regular use of green tea, a plant known to be rich in polyphenols and other phytochemicals [105].

Regarding the action of polyphenols on Sirt1, resveratrol has been identified as an important polyphenol with antiageing effects that acts through the regulation of Sirt1 [61,106,107]. It has been shown that modulation of Sirt1 by resveratrol can affect longevity-related mechanisms [107,108]. In a study on a model of Alzheimer’s disease using senescence-accelerated mouse proper 8 (SAMP8) mice, resveratrol supplementation showed beneficial effects by activating the AMPK pathway and Sirt1, inducing increased cell survival and longevity. Similarly, the neuroprotective effects of resveratrol have been reported when assessing Alzheimer’s disease-related features as an age-related disease [109]. Furthermore, a recent study showed that the antioxidant properties of resveratrol were able to attenuate H_2_O_2_-induced senescence in bone marrow mesenchymal stem cells (BMMSCs), partially via modulation of homologue A of the avian v-relreticuloendotheliosis viral oncogene (RELA) and Sirt1 [62]. Table 1 summarizes the main effect of polyphenols in this hallmark. Thus, the antioxidant and anti-inflammatory properties of polyphenols endow these compounds with an important antiageing capacity, which positively impacts telomere length and prevents telomere shortening, resulting in the prevention of age-related diseases (Figure 3).

## 5. Effects of Polyphenols on Proteostasis Loss

Continuous exposure to several stressors, such as oxidative stress, has allowed cells to develop sophisticated and efficient protection mechanisms, among which the molecular activation of factor 2 related to nuclear factor E2 (Nrf2) plays a fundamental role. Nrf2 is considered to be the master regulator of the antioxidant response and is a critical mechanism for the maintenance of cellular homeostasis and survival [110]. The binding of Nrf2 with the antioxidant response element (ARE) in the regulatory region of many genes leads to the expression of several enzymes with antioxidant and detoxification capabilities. Nrf2 and its invertebrate homologues have emerged as master regulators of cellular detoxification responses and redox status. These stress-sensitive transcription factors function both in acute challenge and as regulators of early antioxidant activity. In the oxidative stress theory of ageing, it is postulated that oxidative damage to biological macromolecules is a key factor in ageing [111,112,113]. Thus, mechanisms that slow down the accumulation of oxidation products in the cells and tissues of an organism may promote longevity. Given the established role of Nrf2 and its invertebrate homologues as master regulators of antioxidant gene expression, studies point to the existence of a strong association between Nrf2 pathway activity and lifespan extension [111,114,115].

Studies performed in both *C. elegans* and *D. melanogaster* showed that activation of Nrf2 provided a significant increase in longevity [116]. It was shown that under normoxic conditions, Nrf2 levels are low, mainly due to its binding to the negative regulator Kelch-like ECH-associated protein 1 (KEAP1), which facilitates Nrf2 ubiquitination and proteasomal degradation [117]. However, during increased oxidative stress, oxidative modification of the cysteine molecule of KEAP1 changes its conformation, resulting in a weakening in its binding to Nrf2 and consequent dissociation. Nrf2, in turn, is no longer subject to degradation and translocates to the nucleus, where it binds to ARE, inducing the expression of cytoprotective genes such as NAD(P)H quinone oxidoreductase 1, glutathione S-transferase, and glutathione reductase. The Keap1-Nrf2/ARE signalling pathway is an important cellular defence mechanism against oxidative damage that simultaneously controls the reduction inreactive oxygen species (ROS) production and oxidative stress while reducing cysteines in Keap1 and subsequently restoring the basal balance of Nrf2 activity [118,119,120].

Several studies have shown that polyphenols can induce Nrf2 activation in different models [42,121,122]. Indeed, in experimental mouse models, the use of resveratrol improved renal function by activating the Nrf2 and Sirt1 signalling pathways and decreasing oxidative stress and mitochondrial dysfunction. In endothelial cells, resveratrol has demonstrated anti-inflammatory effects that appear to be mediated through the induction of Nrf2 [42,116]. Additionally, Nrf2 activation has been shown to increase proteasome activity. Thus, Nrf2 activation increases proteasome expression and activity in an Nrf2-dependent manner, allowing cells to control protein levels by regulated degradation [18,66].

Another mechanism involved in triggering senescence involves the impairment of protein homeostasis or proteostasis [123,124]. Such consequences are attributed, in part, to an accumulation of errors in translation or molecular reading, splicing, and an increase in the production of oxidized and carbonylated proteins, and regulatory systems are therefore needed to preserve the normal functioning of the cellular machinery [123,124,125,126]. The proteostasis network comprises a set of quality control systems involving protein purification mechanisms that allow for the accumulation of damaged proteins and the resulting toxicity to be contained. Ensuring the maintenance of cellular proteostasis demands tight regulation of protein synthesis, folding, conformational maintenance, and degradation. To this end, of the main proteostasis control mechanisms, molecular chaperones of various classes, and their regulators play a key role, coordinating a series of complex and adaptive events [123]. Regarding the systems for removing damaged proteins, the ubiquitin–proteasomal system (UPS), the central proteolytic machinery of mammalian cells, is primarily responsible for proteostasis, as well as the autophagy–lysosomal system, which mediates the renewal of cellular organelles and large aggregates. Many age-related pathologies and the ageing process itself are accompanied by dysregulation of UPS, autophagy, and crosstalk between both systems [123,124]. Failures in the proteostasis system that prevent the degradation of misfolded proteins result in the accumulation and aggregation of these proteins, inducing ageing [127,128]. Although protein quality control networks ensure proteostasis under basal conditions, adverse conditions such as oxidative stress or temperature increases result in conformational stress, causing many additional proteins to become prone to misfolding, particularly affecting protein subtypes that are more vulnerable to instability [123,129].

The cellular maintenance of the functionality and proteostasis requires that proteins irreversibly damaged by oxidation will be degraded and replaced by others from new synthesis. The structural modification and/or damage of proteins by oxidative stress results in a completely dysfunctional, misfolded, insoluble structure, and depending on the level of damage, can even make this molecule resistant to the action of proteases [124]. Under such circumstances, to neutralize oxidative damage to cellular structures, there are effective systems that can be activated to promote an increase in cellular antioxidant status in situations where the basal levels of ROS produced during normal cellular activities are exceeded. If the enzymatic system for eliminating cellular ROS is not sufficient to prevent a redox imbalance, molecular redox sensors, such as Keap1/Nrf2, can be activated very rapidly [124].

Some studies have reported that polyphenols may increase the effectiveness of the proteostasis system by stimulating the proteasome, thereby inducing degradation of damaged proteins, autophagy, and control of oxidative stress [12,130]. It is known that these compounds are effective in multiple molecular mechanisms; however, their activity in the regulation of protein degradation pathways at different stages may be an effective strategy to stop the accumulation of misfolded proteins responsible for the genesis of several diseases [131,132] (Figure 4). Table 1 summarizes the main effect of polyphenols in this hallmark.

## 6. Effects of Polyphenols on Deregulated Nutrient-Sensing Pathways

Polyphenol intake has been shown to be effective in preventing chronic diseases such as heart disease, obesity, cancer, and neurodegenerative diseases. Several investigations attribute these benefits to increased Sirt-1 activity. Sirt-1 is part of the sirtuin family, a class of nutrient-sensitive regulators of epigenetic information capable of modulating senescence and cell lifespan, and is therefore associated with longevity [133,134,135,136,137]. Sirt1 is an NAD^+^-dependent deacetylase and targets several transcription factors involved in adaptive responses to cellular stress, such as NF-κB, forkhead transcription factor (FOXO) 1, 3 and 4, peroxisome proliferator-activated receptor gamma coactivator 1 (PGC-1) and p53. For example, Sirt-1 can inhibit the transcription of p53 through mechanisms of deacetylation of this molecular target, resulting in modulation of pathways involved in the control of cellular ageing [138]. Several studies have shown that sirtuins are strongly influenced by environmental factors, such as dietary habits, which directly affect the regulation of multiple molecular events responsible for the regulation of gene expression, metabolism, DNA repair, apoptosis, and ageing. Activation of this pathway may increase life expectancy [139,140,141].

Recent studies have shown the beneficial effects of polyphenols such as resveratrol, quercetin, curcumin, tannins, and catechins on sirtuin activity [134,142,143]. Indeed, it has been shown that the use of resveratrol induced an increase in the activity of signalling pathways mediated by Sirt-1 and promoted improvements in the brain health of rats. The mechanisms described for such effects include modulation of the inflammatory process, autophagy, increased antioxidant activity, inhibition of apoptosis, promotion of improved plasticity of synaptic pathways, and increased cerebral blood flow [144]. Moreover, it was demonstrated that administration of polyphenols such as quercetin, naringenin, and silymarin was able to reverse the age-related impairment of monoaminergic neurotransmitter secretion by increasing Sirt-1 levels and inhibiting NF-κB in the hippocampus of rats, restoring cognitive functions and motor coordination [145]. Similarly, chronic treatment with catechins induced increased levels of Sirt-1 in the hippocampus of aged rats, resulting in improved cognitive abilities in this model [146]. Table 1 summarizes the main effect of polyphenols in this hallmark. Such findings support the potential of polyphenols to regulate one more important hallmark of ageing, demonstrating that these compounds can substantially contribute to the control of deregulated nutrient sensing pathways by modulating key mechanisms such as Sirt-1 (Figure 5).

## 7. Effects of Polyphenols on Mitochondrial Dysfunction

Ageing leads to a significant reduction in the efficiency of mitochondria in the production of ATP, which results in an increase in free radicals as well as the traffic of these elements through the membranes of these organelles, damaging several cellular structures. These changes culminate in increased oxidative stress and reduced energy production and characterize the establishment of mitochondrial dysfunction [147,148]. There is strong evidence in the literature indicating that the accumulation of oxidative damage in mtDNA during natural ageing is a risk factor for the development of age-related neurodegenerative disorders [149]. It is estimated that the frequency of point mutations in mtDNA increases approximately 5-fold during an 80-year lifespan, indicating that senescence is responsible for promoting profound metabolic and bioenergetic modifications [150,151,152]. Therefore, the removal of dysfunctional mitochondria by mechanisms such as mitophagy is critical for cell survival and health, especially for nervous system cells, because these processes are particularly susceptible to imbalances arising from ageing [153,154]. Furthermore, there is evidence in the literature indicating a strong association between mtDNA dysfunction and telomere length shortening, suggesting common molecular mechanisms and a complex telomere–mitochondria interaction during ageing in humans [5,93,155].

Studies concerning the potential of polyphenols in the modulation of mitochondrial function point to the ability of polyphenol compounds (such as resveratrol, curcumin, hydroxytyrosol and oleuropein) to stimulate mechanisms involved in mitophagy, favouring quality control and mitochondrial clearance [156,157,158,159,160]. Several studies have shown that polyphenols, such as resveratrol, have the ability to induce the expression of PGC-1α and mitochondrial transcription factor A (mtTFA), stimulating mitochondrial biogenesis, and increasing the expression of proteins involved in the control of mitochondrial fission/fusion, thus preserving mitochondrial homeostasis [63,156,161,162,163,164].

These polyphenolic compounds also have antioxidant capabilities, giving these molecules a very important biological role, because oxidative stress is considered to be one of the main mediators of mitochondrial damage and mitophagy impairment that occurs during ageing [165,166,167,168]. Table 1 summarizes the main effect of polyphenols in this hallmark. Therefore, the efficient elimination of nonfunctional organelles and aggregated proteins is fundamental to avoid increased stress and cellular degeneration. The protective action of antioxidants, ROS scavengers, and mitophagy stimulators appears to be the central mechanism for longevity and reduced risk of degenerative diseases [93,153,166,169,170] (Figure 6).

## 8. Effects of Polyphenols on Cell Senescence

Another marker of ageing caused by stress or excessive cellular damage is senescence or cellular ageing. This event is characterized by irreversible cell cycle arrest in the G1 phase. Such mechanism is necessary to restrict replication of old and damaged cells and other harmful changes, thus inactivating potential malignant transformation [171,172,173]. Acute senescence comprises normal biological processes intrinsic to physiological homeostasis and exerts beneficial effects on tissues during embryonic development, wound healing, or tissue repair. However, in extreme situations, this process tends to become chronic, resulting in damaging effects to cells and tissues, particularly in elderly individuals, as these structures lose their ability to remove damaged cells through autophagy, leading to ageing, as well as an increased risk of cancer and other chronic diseases associated with ageing [174]. The accumulation of senescent cells has been shown to be associated with the development of age-related diseases both in humans and other species, such as rodents and primates [175,176,177,178]. Furthermore, evidence points to oxidative stress as one of the main causes of cellular senescence [179,180,181].

Considering the well-known efficacy of polyphenols as potent antioxidant agents, speculations have emerged about the ability of these substances to prevent cellular senescence and delay the ageing process. In this sense, a combined treatment with the senolytic drugs dasatinib and quercetin suppressed the activity of β-galactosidase associated with senescence and resulted in reduced accumulation of senescent cells in human adipose tissue [53]. Based on these findings, a subsequent in vitro study showed that the combination of dasatinib and quercetin was able to attenuate senescence-related idiopathic pulmonary fibrosis [54]. In another study, it was shown that chronic treatment with hydroxytyrosol or oleuropein aglycone effectively reduced the number of senescent presenescent human lung cells and neonatal human dermal fibroblasts, as demonstrated by measuring the number of cells positive for β-galactosidase and the expression of p16 protein [47]. Similarly, oleuropein treatment was shown to delay the onset of senescence morphology and extend the lifespan of human embryonic fibroblast IMR90 and WI38 cells by approximately 15% [50]. The SAMP8-senescence mouse model was used to evaluate the benefits of olive oil polyphenol supplementation. The study showed that a diet enriched with higher levels of olive oil polyphenols was able to significantly reduce oxidative damage in cardiac tissue and induce the expression of longevity-related genes compared to a group that consumed a diet with a lower dosage of olive oil polyphenols [182]. Additionally, a recent study using gallic acid showed that this compound is able to suppress β-galactosidase activity as well as the expression of oxidative stress markers in rat embryonic fibroblast cells [46]. Table 1 summarizes the main effect of polyphenols in this hallmark. These findings suggest that polyphenols may be able to control cellular senescence and thus slow down the ageing process (Figure 7).

## 9. Effects of Polyphenols on Stem Cell Exhaustion

There is evidence showing that a diversity of causal factors can significantly contribute to the functional decline and exhaustion of stem cells (SCs), inducing apoptosis or senescence, as well as compromising the capacity for self-renewal and the cellular and tissue regenerative potential. These factors may stem from intracellular imbalances such as DNA damage, bioenergetic and mitochondrial imbalances, accumulation of poorly bound proteins, and increased levels of ROS caused by the spill over of electrons from the oxidative phosphorylation cascade and may be determined by extracellular factors such as local and/or systemic imbalances and changes in SC niches. Such conditions result in a decreased asymmetric SC division rate, which leads to the exhaustion of progenitor cells and consequently to ageing and age-related diseases [183].

Data from previous studies show that dietary habits can actively control all factors involved in the genesis of SC exhaustion. For example, it has been observed that olive oil consumption may be beneficial in regulating virtually all events involved in the progression of ageing, including SC exhaustion. These benefits have been attributed in part to the significant presence of monounsaturated fatty acids, as well as to other bioactive compounds, including polyphenols such as caffeic acid, tyrosol, hydroxytyrosol, oleocanthal, and oleuropein [184,185]. Furthermore, oleuropein has been shown to stimulate osteoblastogenesis while inhibiting adipogenesis, promoting an osteoblastic phenotype rather than adipocyte differentiation into human bone marrow mesenchymal stem cell (MSC) progenitors [51]. Based on the abovementioned evidence, olive oil consumption has been associated with slower skeletal ageing [186]. Similarly, data obtained with the use of epigenin, another component found in olive oil, showed that this substance was also able to simultaneously modulate the events of osteoblastogenesis and adipogenesis, favouring osteoblast differentiation while inhibiting the transition from preadipocytes to adipocytes [40]. In support of these findings, other studies have shown that olive oil supplementation was able to prevent the decline in bone mineral density and osteoporosis in an in vivo study using ovariectomized rats and in a clinical trial with women undergoing artificial menopause. In addition, olive phytochemicals such as oleuropein and other polyphenols have also been shown to have age-protective effects on haematopoietic progenitor cells by prolonging their lifespan and stimulating asymmetric divisions [41,187]. Oleic acid, in turn, can also induce the secretion of angiogenic factors by MSCs, in addition to promoting tissue regeneration [49,188]. Additionally, other disorders related to SC exhaustion related to senility include the activity of endothelial precursor cells (EPCs), which, under homeostatic conditions, play an important role in the regeneration of vascular endothelium lesions and in the neovascularization of ischaemic tissues through stimulation by angiotensin II. Faced with imbalances, however, the same angiotensin II acts in an antagonistic manner to induce senescence of EPCs, as observed in the pathogenesis of hypertension. In this sense, polyphenols such as oleuropeurin and oleacein were able to exert a protective effect against senescence of SPCs induced by angiotensin II [48]. Because olive oil is an important part of the Mediterranean diet, it is interesting to highlight that this diet, in general, is associated with better health conditions during ageing [189]. The Mediterranean diet is rich in a variety of bioactive compounds, including a variety of vitamins and minerals, fibre, polyphenols, monounsaturated fatty acids (MUFAs) and polyunsaturated fatty acids (PUFAs), many of which have been shown to exert beneficial health effects, both when used individually and in combination [189]. Finally, the Mediterranean diet has also been shown to modulate SC depletion, improving proliferative capacity as well as EPC activity in elderly individuals [186,190,191,192]. Table 1 and Figure 8 summarize the main effect of polyphenols in this hallmark.

## 10. Effects of Polyphenols on Altered Intercellular Communication

In addition to cell autonomous changes, ageing also involves changes at the level of intercellular communication. One prominent ageing-associated change in intercellular communication is inflammageing, which is characterized by a latent proinflammatory phenotype that accompanies ageing in mammals as a result of multiple factors, including accumulation of proinflammatory tissue damage, dysfunctional immune system to effectively kill pathogens and dysfunctional host cells, increased propensity of senescent cells to secrete proinflammatory cytokines (resulting in a senescence-associated secretory phenotype-SASP), activation of the transcription factor NF-κB, and the occurrence of a defective autophagy response [2]. These changes result in increased activation of the NLRP3 inflammasome and other proinflammatory pathways, ultimately leading to increased production of IL-1ß, tumour necrosis factor (TNF), and interferons (IFs) [2]. Furthermore, there is evidence that NF-κB is the major transcription factor involved in regulating the expression of SASP components [193,194,195]. Several studies have highlighted distinct mechanisms involving the activation of innate immunity along with increased proinflammatory mediators and the chronic inflammatory process with ageing [196,197,198,199].

In this sense, the role of polyphenols in modulating the inflammatory cascade is already wellknown [158,196,200,201]. It has been shown that resveratrol can inhibit NF-κB-regulated inflammatory cytokines via regulation of SIRT1 [64]. In addition, another study also stated that resveratrol induced increased expression of PPAR-γ and SIRT1, resulting in decreased inflammatory status [202]. Moreover, the activation of SIRT1 by resveratrol can also inhibit TNF-α-induced IL-1β and IL-6 expression and reduce the phosphorylation of rapamycin (mTOR) and S6 ribosomal protein (S6RP) in a fibroblast cell line (3T3) [65]. In aged mice, resveratrol treatment was able to reduce IL-1β and TNF-α mRNA levels. Additionally, reduced levels of ASC (apoptosis-associated protein-like component of the NLRP3 inflammasome containing a CARD), caspase-1, and NALP-3 (protein 3 containing NACHT, LRR and PYD domains) were observed in this model [203]. Resveratrol treatment also reduced the expression of IL-1β and TNF-α and inhibited stroke-induced brain damage and inflammation in aged female mouse models [204].

Quercetin is another widely studied polyphenol that has recognized anti-inflammatory properties [196,205,206,207]. In vitro research performed with quercetin has shown that this polyphenol was able to significantly reduce TNF-α expression in murine glial cells and macrophages after induction with LPS [55,56]. Furthermore, in a study performed on A549 lung cells, quercetin inhibited IL-8 [57]. In another study, it was shown that this polyphenol was able to significantly inhibit the selective release of IL-6 stimulated by IL-1 in mast cells [58].

Curcumin has been used as a traditional medicine in India and other Asian countries for many diseases [196,208]. The antioxidant, anti-inflammatory, and anticancer properties of curcumin have made it a potential nutritional supplement [208,209,210,211]. The anti-inflammatory role of curcumin has been well established in recent decades [196,211,212,213,214,215,216]. For instance, it has been reported that curcumin can potentially reduce the activity of several inflammation-related transcription factors, including NF-κB, AP-1, signal transducer and activator of transcription (STAT), and hypoxia-inducible factor-1 (HIF-1). Furthermore, it was found that inhibition of NF-κB activity by curcumin resulted in subsequent inhibition of P65 translocation to the nucleus and suppression of IκB-α degradation [217]. Further studies have shown that curcumin-induced inhibition of the transcription factor NF-κB results in a reduction in TNF-α, IL-6 and COX-2 [43,218,219].

EGCG has several beneficial properties for human health, attributed to its antioxidant, anti-inflammatory, and anticancer properties [220,221,222,223,224,225,226]. EGCG has been shown to exert its anti-inflammatory effects through inhibition of TNF-α, COX-2, and iNOS expression [44]. Furthermore, EGCG can directly suppress NF-κB and AP-1 in human ECV304 cells [45]. Table 1 and Figure 9 summarize the main effect of polyphenols in this hallmark.

In summary, polyphenols have demonstrated abilities to act on multiple targets associated with chronic inflammation, proving to be useful against inflammatory and inflammation-associated diseases. Taken together, these data point to a potential anti-inflammageing of bioactive compounds, appearing as a promising resource for the prevention of inflammatory diseases associated with ageing.

## 11. Polyphenols’ Regulation of Ageing—A Clinical Perspective

We searched on Clinical Trials (clinicaltrials.gov) for the terms that are the subject of this review. In Table 2, we summarize the completed clinical trials conducted in elderly subjects in which the effects of polyphenols have been evaluated.

The effects of green tea supplementation were evaluated in a nonblinded, non-placebo-controlled study design (NCT01594086). Although the study had a number of limitations, the authors described that three months of green tea consumption improved cognitive dysfunction in the elderly [227]. COSMOS-Mind was the first large-scale, long-term randomized controlled trial (NCT03035201) to assess the long-term effects of cocoa extract (containing 500 mg/day of cocoa flavanols) on global cognition in older women and men. However, the authors stated that daily ingestion of cocoa extract for 3 years had no impact on cognition [228]. Subsequently, another large clinical trial was conducted in older adults (COSMOS Web-NCT04582617). This study aimed to evaluate the effects of a dietary intervention with cocoa extract on cognitive changes and brain structure and function over a 2-year period. Although the status of the study is complete as of December 2022, no results are publicly available.

Recently the effects of curcumin supplementation were also evaluated in older adults. In a double-blind, placebo-controlled, crossover design study (NCT04119752), curcumin was found to improve brain oxygenation and blood volume during exercise, highlighting the potential of this compound [229]. Additionally, a phase 2 clinical trial (NCT03085680) showed that curcumin improves cognitive and physical function in older adults.

The effects of resveratrol supplementation in older adults have also been evaluated in clinical trials. Initially, a double-blind, randomized, placebo-controlled trial (NCT01126229) was designed to evaluate safety and metabolic outcomes after 3 months of resveratrol use at doses of 300 mg and 1000 mg. The results of this work indicate that resveratrol is well tolerated in overweight, older adults, a population known to be at high risk for chronic diseases [230]. Subsequently, a three-arm, two-site pilot randomized controlled trial (NCT02523274) indicated that exercise combined with resveratrol is safe and feasible for older adults with functional limitations. In addition, exercise combined with resveratrol 1000 mg/d increases mobility-related indices of physical activity and mitochondrial function [231]. Certainly, if polyphenols are well tolerated and safe for future investigations, clinical trials targeting ageing, especially phase 2 and 3 studies with larger populations, are urgently needed for better management and care of elderly people.

## 12. Conclusions

In general, a plethora of evidence highlights the effectiveness of polyphenolic compounds in delaying the ageing process and protecting against the development of diseases related to this phenomenon in several models. Polyphenols can be found mainly in foods and beverages of plant origin, such as fruits, vegetables, and legumes, as well as in herbs such as teas and spices; therefore, the most recommended and affordable way to acquire these phytochemicals is to eat polyphenol-rich foods as part of a regular diet. The consumption of polyphenol-rich foods provides numerous benefits, attributed mainly to their antioxidant properties, with a consequent reduction inoxidative stress, strongly associated with tissue ageing. In addition, several studies have reported immunomodulatory, anti-inflammatory, and anticancer effects, among others, conferred by these bioactive compounds. Such properties have been shown to influence the preservation of telomere length, contributing to increased longevity, as illustrated in Figure 10. These bioactives have also benefited the function of stem cells, favouring tissue regeneration, and a beneficial association has been demonstrated between the consumption of dietary polyphenols and the preservation of cognitive functions during ageing, exalting their neuroprotective potential.

Thus, the beneficial action of polyphenols on ageing markers is clear. These data, in general, indicate that these properties may be useful in the development of therapeutic strategies directed at the development and maintenance of health, reducing the risk of diseases associated with ageing and increasing longevity.

## Figures and Tables

**Figure 1 ijms-24-05508-f001:**
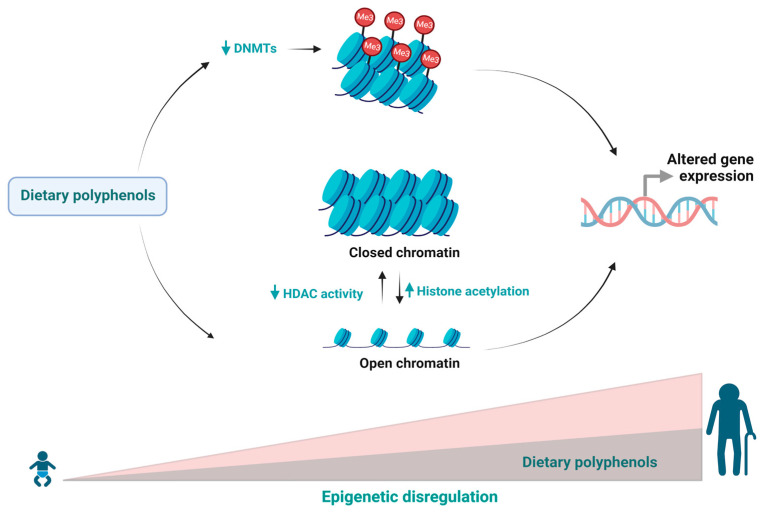
During ageing, epigenetic regulation changes in response to exogenous and endogenous factors affecting various biological processes. Dietary polyphenols can regulate epigenetic mechanisms and delaying the ageing process.

**Figure 2 ijms-24-05508-f002:**
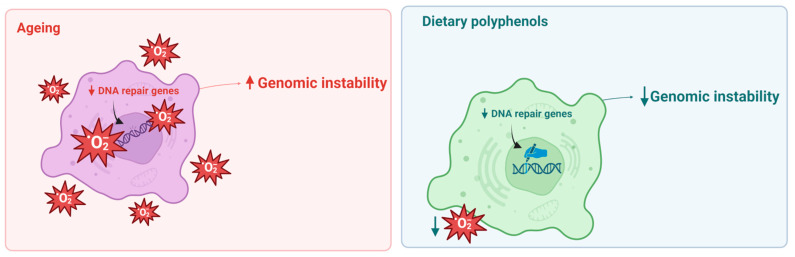
Genome instability is an important hallmark of ageing (**left** panel). Dietary polyphenols can decrease genomic instability through the regulation of ROS levels and increase DNA repair activity (**right** panel).

**Figure 3 ijms-24-05508-f003:**
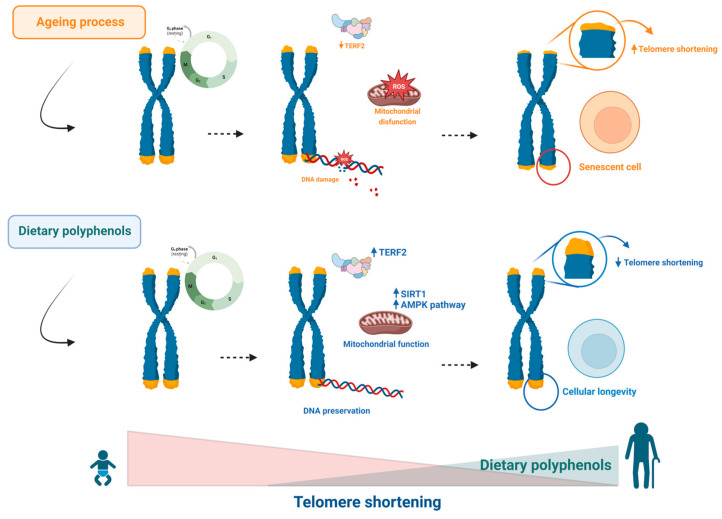
Telomere shortening contributes to ageing. Short telomeres lead to mitochondrial DNA damage, inducing senescence. Dietary polyphenols can protect against ageing by modulating each step in this sequence.

**Figure 4 ijms-24-05508-f004:**
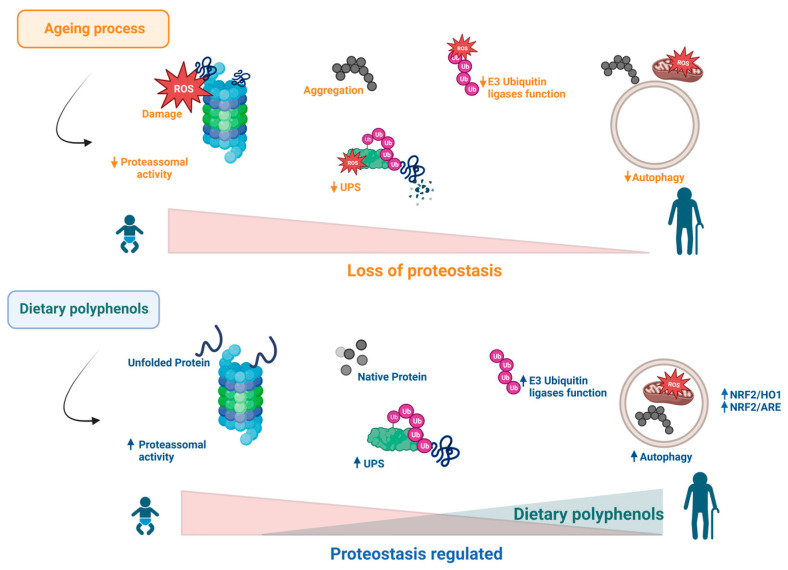
During ageing, the increase in ROS affects proteasomal and autophagy processes. Consequently, proteostasis is lost, as shown by the detrimental accumulation of protein aggregates and damaged organelles such as mitochondria. Several reports have been shown that dietary polyphenols can reverse the deleterious effects associated with this hallmark of ageing.

**Figure 5 ijms-24-05508-f005:**
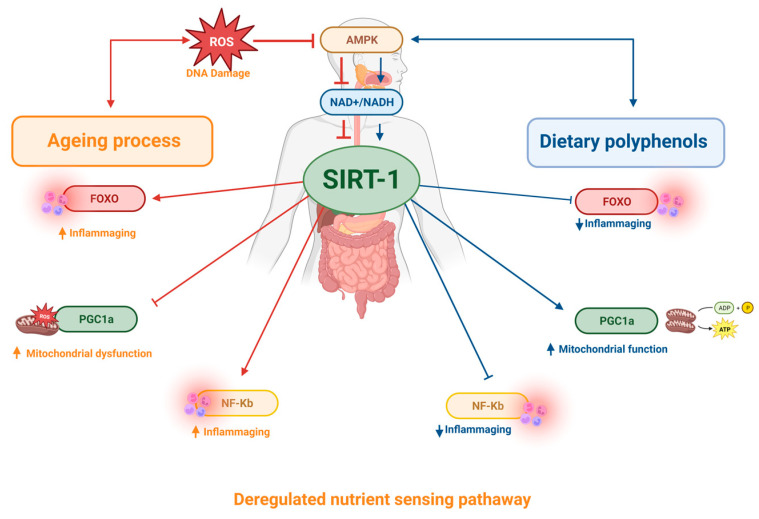
During ageing, in response to cellular stress, SIRT1 activity is compromised, resulting in loss of sensitivity to detect nutrients and activation of inflammation factors. Dietary polyphenol favours the activation of SIRT1, modulating senescence and cell lifespan, and therefore is associated with longevity.

**Figure 6 ijms-24-05508-f006:**
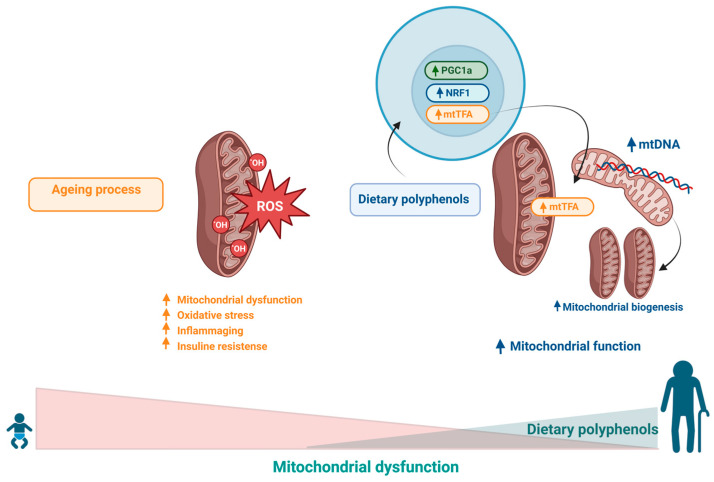
Oxidative damage to mtDNA results from the accumulation of free radicals within mitochondria in response to the damage involved in the ageing process, which culminates in increased oxidative stress and reduced energy production. Dietary polyphenols could improve mitochondrial function through the expression of PGC-1α and mitochondrial transcription factor A (mtTFA), stimulating mitochondrial biogenesis.

**Figure 7 ijms-24-05508-f007:**
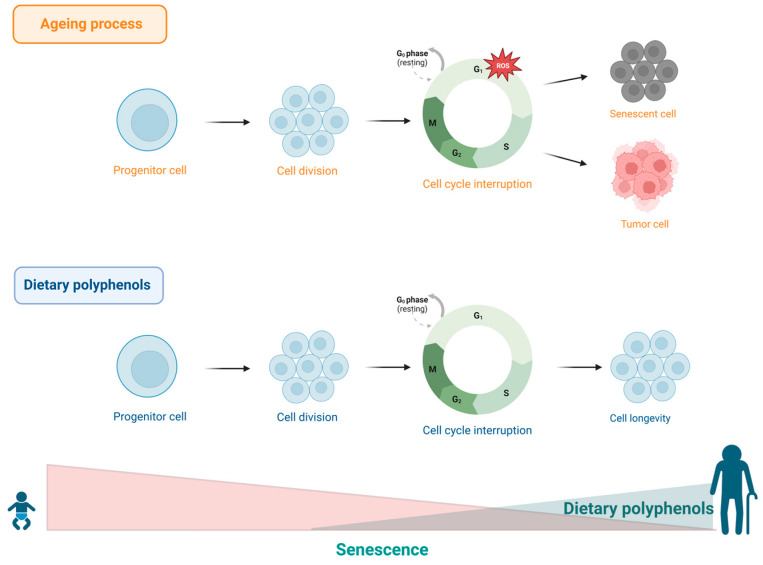
During ageing, stress or excessive cellular damage favours cellular ageing. This process is characterized by irreversible arrest of the cell cycle in the G1 phase. Dietary polyphenols offer benefits to cellular homeostasis, contributing to cell longevity.

**Figure 8 ijms-24-05508-f008:**
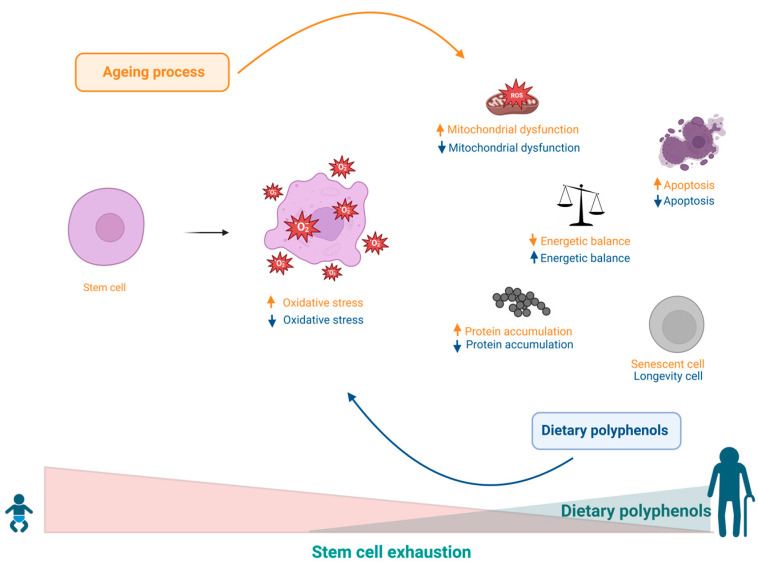
Increased ROS levels caused by electron spillage from the oxidative phosphorylation cascade cause local and/or systemic imbalances and changes in stem cell niches, resulting in a decreased asymmetric stem cell division rate, leading to depletion of progenitor cells, and consequently contributing to senescence. Polyphenols can benefit the regulation of virtually all events involved in the progression of ageing.

**Figure 9 ijms-24-05508-f009:**
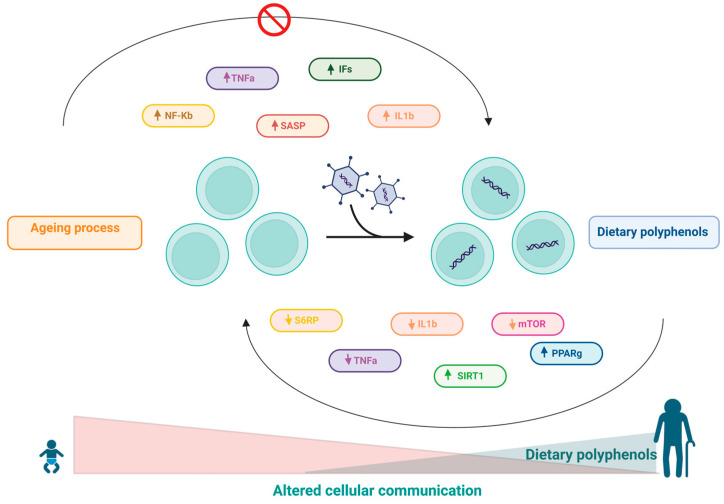
Inflammageing results from autonomous changes in cells, which involve changes in cell communication during the ageing process and includes accumulation of proinflammatory tissue damage. Apolyphenol-based diet can modulate the inflammatory cascade through the regulation of the SIRT1 pathway, resulting in a decrease in the inflammatory state.

**Figure 10 ijms-24-05508-f010:**
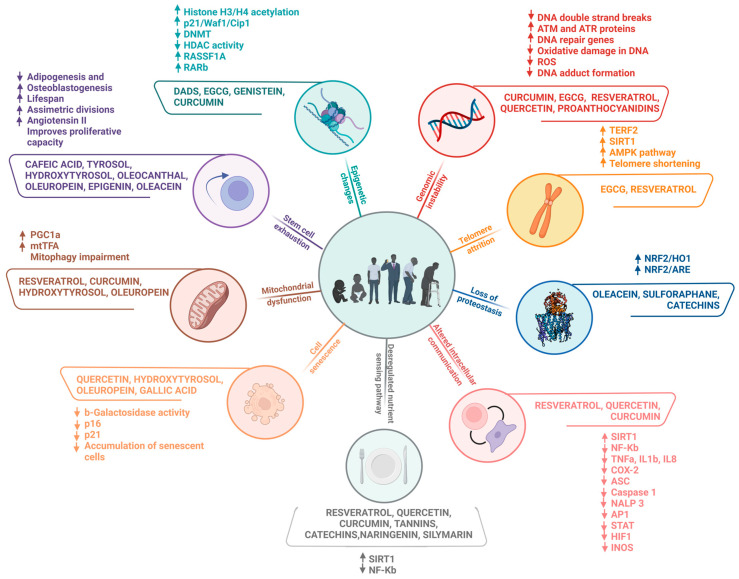
Summary of the main effects of polyphenolic compounds on the molecular aspects involved in ageing hallmarks.

**Table 1 ijms-24-05508-t001:** The main effects of polyphenols on ageing hallmarks.

Polyphenols	Models	Main Effects	Ref
Apigenin	MSCs Hematopoietic stem cell	↓Adipogenesis/↑osteoblastogenesis ↑Lifespan ↑Asymmetric divisions	[40] [41]
Catechins	Hepa1c1c7 cells	↑ARE/Nrf2	[42]
Curcumin	MCF-7 cells A549 cells Microglial cells Astrocytes cells	↓DNMT1; ↑RASSF1A ↓DNMT3b; ↑RARβ ↓NF-κB ↓TNF-α ↓IL-6 ↓COX-2	[36] [38] [39] [43]
DADS	Caco-2 cells HT-29 cells	↑HDAC activity; ↑H3 and H4 acetylation; ↑p21waf1/cip1	[30]
EGCG	A549 cells HL60 cells KYSE-150 cells PC3 cells MCF-7cells Macrophages ECV304 cells	↓Cell proliferation; ↑Cell cycle arrest in G1 phase; ↑Apoptotic activity; ↓hTERTmethylation; ↓H3K9 acetylation; ↓DNMT1, DNMT3a and DNMT3b; ↓HDAC activity. ↓NF-κB ↓TNF-α ↓iNOS ↓COX-2 ↓AP-1	[31] [32] [33] [34] [44] [45]
Gallic acid	Rat embryonic fibroblast cells	↓β-galactosidase activity ↓ROS	[46]
Hydroxytyrosol	MRC5 cells NHDF cells	↓NFκB ↓β-galactosidadse activity ↓Accumulation of senescent cells ↓p16 ↓SASP (IL-6, COX2, TNFα)	[47]
Oleacein	SPCs	↑ Angiotensin II ↑Nrf2/heme oxygenase-1	[48]
Oleic acid	MSCs	↑EphB2	[49]
Oleuropein	MRC5 cells NHDF cells IMR90 cells WI38 cells MSCs Hematopoietic stem cell SPCs	↓NFκB ↓β-galactosidadse activity ↓Accumulation of senescent cells ↓p16 ↓SASP(IL-6, COX2, TNFα) ↓ROS ↑Proteasome activity ↓Adipogenesis/↑osteoblastogenesis ↑Lifespan ↑AsyMmetricdivisions ↑Angiotensin II ↑Nrf2/heme oxygenase-1	[47] [50] [51] [41] [48]
Procyanidins	BEAS-2B WRL-68	↓DNA damage ↑ATM and ATR activity	[52]
Quercetin	Primary human fibroblasts HUVEC cultures Murine glial cells Macrophages A549 Mast cells	↓SASP ↓β-galactosidadse activity ↓Accumulation of senescent cells ↓p16 ↓p21 ↓TNF-α ↓IL-8	[53] [54] [55] [56] [57] [58]
Resveratrol	Mouse embryonic fibroblasts IMR-90 cells BMMSCs Rat cardiomyocyte cultures 3T3	↓Mutations in the ARF/p53 pathway ↓DNA double-strand breaks ↑Telomerase activity ↑SIRT1 ↑Sirt1/Sirt3-FoxO pathway ↑Sirt1/Sirt3-Mfn2-Parkin-PGC-1α Pathway ↓NF-κB ↓TNF-α, IL-1β and IL-6 expression ↓mTOR phosphorylation) ↓S6RP	[59] [60] [61] [62] [63] [64] [65]
Sulphoraphane	Embryonic fibroblast cells	↑Nrf2/ARE	[66]

↑—induction; ↓—repression.

**Table 2 ijms-24-05508-t002:** Completed clinical trials evaluating the effects of selected polyphenols on ageing.

Treatment	Trial ID	Phase	N^o^ Pts	Condition	Results	Ref
Green tea powder 2 g/day for 3 months	NCT01594086	NA	15	Cognitive dysfunction in elderly	Improved cognitive function and reduced progression of vascular dementia	[227]
Cocoa extract (500 mg/d flavanols, including 80 mg. (-)-epicatechins) 3-year trial	NCT03035201	NA	2262	Cognitive dysfunction in elderly	Cocoa extract did not benefit cognition	[228]
Cocoa extract (500 mg/d flavanols, including 80 mg. (-)-epicatechins) 2-year trial	NCT04582617	NA	4000	Cognitive changes and brain structure	No results posted	
10 g of curcumin supplementation	NCT04119752	NA	28	Elderly with cardiometabolic risks	Improved cerebral oxygenation and blood volume	[229]
Curcumin (1000 mg/day) 3-month trial	NCT03085680	Phase 2	17	Ageing adults at increased risk for disability	Improvement in attention, memory, and physical function grip strength	
300 or 1000 mg/d of resveratrol, 3-month trial	NCT01126229	Phase 1	32	Overweight, older adults	Supports the safety of resveratrol supplementation in this at-risk population	[230]
500 or 1000 mg of resveratrol per day, 3-month trial	NCT02523274	Phase 2	60	Ageing	Improved skeletal muscle mitochondrial function and mobility-related indices of physical function	[231]

NA—not applicable.

## Data Availability

Not applicable.

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
