# Peer review of "The Molecular Mechanism of Polyphenols in the Regulation of Ageing Hallmarks"

_ijms, 2023, doi:10.3390/ijms24065508_

Round 1

Reviewer 1 Report

Overall, the review "The molecular mechanism of polyphenols in the regulation of 2 ageing hallmarks” by Pereira et al is well written, well organized and manages to express the concept successfully. The English used is excellent, thus there are no corrections or comments to make. Images and tables are very clear and well-constructed.

 This review summarizes the knowledge on the benefits of polyphenols on the ageing process, as well as the main molecular mechanisms responsible for their antiaging effects.

The only comments about the review are:

·      - From line 347 to 356 the negative effects of the excessive use of polyphenols were discussed. Considering that this is an essential element when discussing the subject, would it not be more effective to quote this paragraph directly in the introduction? This would add further information for the reader when introducing the topic;

·      - Many references of in vivo and in vitro studies are cited throughout the text. Are there also studies at the stage of clinical trials on some of the substances mentioned? If so, it would be interesting to discuss it briefly in the text;

·     - Several times through the text "in vivo" or "in vitro" are written in Roman type. It would be more correct to use italics.

Author Response

Reviewer 1Overall, the review "The molecular mechanism of polyphenols in the regulation of  ageing hallmarks” by Pereira et al is well written, well organized and manages to express the concept successfully. The English used is excellent, thus there are no corrections or comments to make. Images and tables are very clear and well-constructed.

 This review summarizes the knowledge on the benefits of polyphenols on the ageing process, as well as the main molecular mechanisms responsible for their antiaging effects.

The only comments about the review are:

Comment 1) - From line 347 to 356 the negative effects of the excessive use of polyphenols were discussed. Considering that this is an essential element when discussing the subject, would it not be more effective to quote this paragraph directly in the introduction? This would add further information for the reader when introducing the topic; 

 Answer: We acknowledge the referee for his/her comment, and have updated the information on the Introduction (Line 69-78)

 Comment 2)- Many references of in vivo and in vitro studies are cited throughout the text. Are there also studies at the stage of clinical trials on some of the substances mentioned? If so, it would be interesting to discuss it briefly in the text

Answer: We acknowledge the referee for his/her comment, and have added the information on the new section “11. Polyphenols regulation of ageing - a clinical perspective”

Comment 3)-  Several times through the text "in vivo" or "in vitro" are written in Roman type. It would be more correct to use italics.

 Answer: We acknowledge the referee for his/her comment, and have corrected it into the manuscript

Reviewer 2 Report

In this manuscript, general way the main findings described in the literature about the benefits of polyphenols and main regulatory mechanisms responsible for the observed antiaging effects were summarized. The authors introduced several parts: age-related effects of polyphenols on epigenetic changes, genomic instability, telomere attrition, proteostasis loss, deregulated nutrient sensing pathways, mitochondrial dysfunction, cell senescence, stem cell exhaustion and altered intercellular communication. The paper is well organized and illustrated. In my opinion, this paper has high importance and should be accepted by International Journal of Molecular Sciences. I have some comments here:

1.      I suggest that more figures can be added to elucidate the effects of polyphenols and signalling pathways.

2.      Is there some relevance between the concentration of H2S and polyphenols? Because higher concentration of ROS will reduce the concentration of H2S.

Author Response

Reviewer 2In this manuscript, general way the main findings described in the literature about the benefits of polyphenols and main regulatory mechanisms responsible for the observed antiageing effects were summarized. The authors introduced several parts: age-related effects of polyphenols on epigenetic changes, genomic instability, telomere attrition, proteostasis loss, deregulated nutrient sensing pathways, mitochondrial dysfunction, cell senescence, stem cell exhaustion and altered intercellular communication. The paper is well organized and illustrated. In my opinion, this paper has high importance and should be accepted by International Journal of Molecular Sciences. I have some comments here:

Comment 1) I suggest that more figures can be added to elucidate the effects of polyphenols and signalling pathways.

Answer: We acknowledge the referee for his/her comment, and have added a new set of Figures.

  1. Is there some relevance between the concentration of H2S and polyphenols? Because higher concentration of ROS will reduce the concentration of H2S.---

Answer: We agree with the reviewer that polyphenols, due to their antioxidant properties, contribute to maintaining stable concentrations of H2S, contributing to maintaining their biological roles [1-4]. In addition to its relevant biological role in the regulation of numerous physiological responses, for example, anti-inflammatory, oxidative stress, neuromodulation, vasoregulation, protection against reperfusion injury after myocardial infarction, and insulin resistance [5-11], several studies have been reported the involvement of H2S in several phases of cell signaling, cell function and cytoprotection with the use of autoxidized polyphenolics, which lead to the oxidation of H2S to polysulfides and thiosulfates responsible for their cytoprotective effects [1; 12-15]. Considering its complexity, we believe that this subject would be out of the scope of this review

  1. Olson K.R., Briggs A., Devireddy M., Iovino N.A., Skora N.C., Whelan J., Villa B.P., Yuan X., Mannam V., Howard S., et al. Green tea polyphenolic antioxidants oxidize hydrogen sulfide to thiosulfate and polysulfides: A possible new mechanism underpinning their biological action. Redox Biol. 2020;37:101731. doi: 10.1016/j.redox.2020.101731.
  2. Fukuto J.M., Ignarro L.J., Nagy P., Wink D.A., Kevil C.G., Feelisch M., Cortese-Krott M.M., Bianco C.L., Kumagai Y., Hobbs A.J., et al. Biological hydropersulfides and related polysulfides—A new concept and perspective in redox biology. FEBS Lett. 2018;592:2140–2152. doi: 10.1002/1873-3468.13090.
  3. Hourihan J.M., Kenna J.G., Hayes J.D. The gasotransmitter hydrogen sulfide induces nrf2-target genes by inactivating the keap1 ubiquitin ligase substrate adaptor through formation of a disulfide bond between cys-226 and cys-613. Antioxid. Redox Signal. 2013;19:465–481. doi: 10.1089/ars.2012.4944.
  4. Yang G., Zhao K., Ju Y., Mani S., Cao Q., Puukila S., Khaper N., Wu L., Wang R. Hydrogen sulfide protects against cellular senescence via S-sulfhydration of Keap1 and activation of Nrf2. Antioxid. Redox Signal. 2013;18:1906–1919. doi: 10.1089/ars.2012.4645.
  5. Zanardo R.C., Brancaleone V., Distrutti E., Fiorucci S., Cirino G., Wallace J.L. Hydrogen sulfide is an endogenous modulator of leukocyte-mediated inflammation. FASEB J. 2006;20:2118–2120. doi: 10.1096/fj.06-6270fje.
  6. Yonezawa D., Sekiguchi F., Miyamoto M., Taniguchi E., Honjo M., Masuko T., Nishikawa H., Kawabata A. A protective role of hydrogen sulfide against oxidative stress in rat gastric mucosal epithelium. Toxicology. 2007;241:11–18. doi: 10.1016/j.tox.2007.07.020.
  7. Abe K., Kimura H. The possible role of hydrogen sulfide as an endogenous neuromodulator. J. Neurosci. 1996;16:1066–1071. doi: 10.1523/JNEUROSCI.16-03-01066.1996.
  8. Laggner H., Hermann M., Esterbauer H., Muellner M.K., Exner M., Gmeiner B.M., Kapiotis S. The novel gaseous vasorelaxant hydrogen sulphide inhibits angiotensin-converting enzyme activity of endothelial cells. J. Hypertens. 2007;25:2100–2104. doi: 10.1097/HJH.0b013e32829b8fd0.
  9. Sivarajah A., Collino M., Yasin M., Benetti E., Gallicchio M., Mazzon E., Cuzzocrea S., Fantozzi R., Thiemermann C. Anti-apoptotic and anti-inflammatory effects of hydrogen sulfide in a rat model of regional myocardial I/R. Shock. 2009;31:267–274. doi: 10.1097/SHK.0b013e318180ff89.
  10. Zhang H., Huang Y., Chen S., Tang C., Wang G., Du J., Jin J. Hydrogen sulfide regulates insulin secretion and insulin resistance in diabetes mellitus, a new promising target for diabetes mellitus treatment? A review. J. Adv. Res. 2021;27:19–30. doi: 10.1016/j.jare.2020.02.013.
  11. Magli E, Perissutti E, Santagada V, et al. H2S Donors and Their Use in Medicinal Chemistry. Biomolecules. 2021;11(12):1899. Published 2021 Dec 18. doi:10.3390/biom11121899.
  12. Olson KR, Gao Y, Straub KD. Oxidation of Hydrogen Sulfide by Quinones: How Polyphenols Initiate Their Cytoprotective Effects. Int J Mol Sci. 2021;22(2):961. Published 2021 Jan 19. doi:10.3390/ijms22020961
  13. Olson KR, Gao Y, Briggs A, et al. 'Antioxidant' berries, anthocyanins, resveratrol and rosmarinic acid oxidize hydrogen sulfide to polysulfides and thiosulfate: A novel mechanism underlying their biological actions. Free Radic Biol Med. 2021;165:67-78. doi:10.1016/j.freeradbiomed.2021.01.035
  14. Olson KR, Gao Y, Arif F, et al. Metabolism of hydrogen sulfide (H2S) and Production of Reactive Sulfur Species (RSS) by superoxide dismutase. Redox Biol. 2018;15:74-85. doi:10.1016/j.redox.2017.11.009
  15. DeLeon ER, Gao Y, Huang E, Olson KR. Garlic oil polysulfides: H2S- and O2-independent prooxidants in buffer and antioxidants in cells. Am J Physiol Regul Integr Comp Physiol. 2016;310(11):R1212-R1225. doi:10.1152/ajpregu.00061.2016

Reviewer 3 Report

The work presented for review is aimed at some general systematization of information on the importance of using polyphenols and their possible impact on the modulation of the main markers of aging described in the world literature.

The large number of naturally occurring polyphenols present both an enticing opportunity and an inherently significant obstacle to finding new treatments for major risk factors associated with aging. Understanding the mechanisms, modes of action, tissue specificity, timing and extent of polyphenol application in various disease processes has proven to be a difficult but achievable goal. As the scientific literature provides evidence for both preventive and therapeutic benefits, public interest is also growing. Growing awareness that polyphenol-rich diets can directly benefit human health is driving further research interest in these important compounds.

The manuscript is based on a careful review of over 200 references. The presented in-depth data analysis indicates the effectiveness of polyphenolic compounds in delaying the aging process and protecting against the development of diseases associated with this phenomenon in several models. This work is worthy of publication due to the need for further research that may expand our understanding of the dynamic role that these dietary substances play in attenuating some risk factors associated with aging.

Author Response

Reviewer 3The work presented for review is aimed at some general systematization of information on the importance of using polyphenols and their possible impact on the modulation of the main markers of aging described in the world literature.

The large number of naturally occurring polyphenols present both an enticing opportunity and an inherently significant obstacle to finding new treatments for major risk factors associated with aging. Understanding the mechanisms, modes of action, tissue specificity, timing and extent of polyphenol application in various disease processes has proven to be a difficult but achievable goal. As the scientific literature provides evidence for both preventive and therapeutic benefits, public interest is also growing. Growing awareness that polyphenol-rich diets can directly benefit human health is driving further research interest in these important compounds.

The manuscript is based on a careful review of over 200 references. The presented in-depth data analysis indicates the effectiveness of polyphenolic compounds in delaying the aging process and protecting against the development of diseases associated with this phenomenon in several models. This work is worthy of publication due to the need for further research that may expand our understanding of the dynamic role that these dietary substances play in attenuating some risk factors associated with aging.

Answer: We acknowledge the referee for his/her comment.

Reviewer 4 Report

Although in principle the topic is of considerable interest to the scientific community focused on nutraceutical studies, I do not find originality and innovation in the paper. Therefore, some flaws hamper its suitability for publication.

Author Response

Reviewer 4Although in principle the topic is of considerable interest to the scientific community focused on nutraceutical studies, I do not find originality and innovation in the paper. Therefore, some flaws hamper its suitability for publication.

Answer: We acknowledge the referee for his/her comment. In fact, the review itself is not original, but is an extensive update of the literature. The new and improved version of the article is more comprehensive, including human studies. I hope it is suitable for publication.

Reviewer 5 Report

The work is an exhaustive review of the modern scientific literature on the beneficial effects of polyphenols in the aging process of the body. The authors extensively discussed the mechanisms of action of polyphenols on various aging characteristics and the resulting anti-aging effects. The influence of polyphenols on age-related epigenetic changes, on genome instability, on stem cell depletion, on age-related telomere shortening, on mitochondrial dysfunction, and on altered intercellular communication has been described. Consuming foods rich in polyphenols brings numerous benefits, mainly attributed to their antioxidant properties, and consequently the reduction of oxidative stress, strongly associated with tissue aging. In addition, several studies have demonstrated, among other things, the immunomodulatory, anti-inflammatory and anti-cancer effects imparted by these bioactive compounds. It was found that polyphenols can be used to delay the aging process by influencing the modulation of the main aging markers described in the literature.

Author Response

Reviewer 5The work is an exhaustive review of the modern scientific literature on the beneficial effects of polyphenols in the aging process of the body. The authors extensively discussed the mechanisms of action of polyphenols on various aging characteristics and the resulting anti-aging effects. The influence of polyphenols on age-related epigenetic changes, on genome instability, on stem cell depletion, on age-related telomere shortening, on mitochondrial dysfunction, and on altered intercellular communication has been described. Consuming foods rich in polyphenols brings numerous benefits, mainly attributed to their antioxidant properties, and consequently the reduction of oxidative stress, strongly associated with tissue aging. In addition, several studies have demonstrated, among other things, the immunomodulatory, anti-inflammatory and anti-cancer effects imparted by these bioactive compounds. It was found that polyphenols can be used to delay the aging process by influencing the modulation of the main aging markers described in the literature.

Answer: We acknowledge the referee for his/her comment.

Round 2

Reviewer 4 Report

In my opinion, the manuscript does not provide the information that merits publication, especially in the JMS.